# Learning Parities with Neural Networks

**Amit Daniely**
School of Computer Science,
The Hebrew University, Israel.
Google Research Tel Aviv.
`amit.daniely@mail.huji.ac.il`

**Eran Malach**
School of Computer Science,
The Hebrew University, Israel.
`eran.malach@mail.huji.ac.il`

## Abstract

In recent years we see a rapidly growing line of research which shows learnability of various models via common neural network algorithms. Yet, besides a very few outliers, these results show learnability of models that can be learned using *linear* methods. Namely, such results show that learning neural-networks with gradient-descent is competitive with learning a linear classifier on top of a data-independent representation of the examples. This leaves much to be desired, as neural networks are far more successful than linear methods. Furthermore, on the more conceptual level, linear models don't seem to capture the "deepness" of deep networks. In this paper we make a step towards showing leanability of models that are inherently non-linear. We show that under certain distributions, sparse parities are learnable via gradient decent on depth-two network. On the other hand, under the same distributions, these parities *cannot* be learned efficiently by linear methods.

## 1 Introduction

The remarkable success of neural-networks has sparked great theoretical interest in understanding their behavior. Impressively, a large number of papers [4, 26, 10, 8, 6, 15, 11, 20, 2, 3, 7, 28, 25, 13, 21, 5, 7, 16, 19, 18, 9] have established polynomial-time learnability of various models by neural networks algorithms (i.e. gradient based methods). Yet, to the best of our knowledge, with the single exception of learning one neuron [27], all these results prove learnability of *linear models*. Namely, models that can be realized by a linear classifier, on top of a (possibly random) embedding that is fixed and does not depend on the data. This is not surprising, as the majority of these papers prove learnability via "linearization" of the network at the vicinity of the initial random weights.

While these results achieved remarkable progress in understanding neural-networks, they are still disappointing in some sense. Indeed, in practice, neural-networks' performance is far better than linear methods, a fact that is not explained by these works. Moreover, learning a linear classifier on top of a fixed embedding seems to completely miss the "deepness" of deep learning.

How far can neural network theory go beyond linear models? In this work we show a family of distributions on which neural-networks trained with gradient-descent achieve small error. On the other hand, approximating the same family using a linear classifier on top of an embedding of the input space in $\mathbb{R}^N$, requires $N$ which grows exponentially, or otherwise requires a linear classifier with exponential norm. Specifically, we focus on a standard and notoriously difficult family of target functions: parities over small subsets of the input bits. We show that this family is learnable with neural-networks under some specific choice of distributions. This implies that neural-networks algorithms are strictly stronger than linear methods, as the same family cannot be approximated by any polynomial-size linear model.

## 1.1 Related Work

Recently, a few works have provided theoretical results demonstrating that neural-networks are stronger than *random features* - a linear model where the embedding map is randomly drawn from some predefined distribution [22]. These works show problems that are easy to learn with neural-networks, while being hard to learn with random features. The work of [27] shows that random features cannot approximate a distribution generated by a single neuron and Gaussian inputs, which is known to be learnable by a neural-network. The work of [1] shows that neural-networks are more efficient than random features, in terms of sample complexity and run-time, for some regression problems generated by a ResNet-like network. A work by [14] shows other family of distributions where neural-networks with quadratic activation outperform random features, when the number of features is smaller than the dimension.

Our result differs from these works in several aspects. First, [27] and [1] study the power of approximating a regression problems, and hence their results cannot be applied to the setting of classification, which we cover in this work. Second, we give an *exponential separation*, while [1] and [14] only give *polynomial* separation. Namely, the problems for which they show that neural networks perform better than linear methods are still poly-time learnable by linear methods.

## 2 Problem Setting

Let $\mathcal{X} = \left\{ \pm \frac{1}{\sqrt{n}} \right\}^n$ be the instance space, and $\mathcal{Y} = \{\pm 1\}$ the label space. Since we focus on a binary classification task, we take the hinge-loss $\ell(y, \hat{y}) = \max\{1 - y\hat{y}, 0\}$ to be our primary loss function. For some distribution $\mathcal{D}$ over $\mathcal{X} \times \mathcal{Y}$, and some function $h : \mathcal{X} \to \mathcal{Y}$, we define the loss of $h$ over the distribution to be:

$$L_{\mathcal{D}}(h) = \mathbb{E}_{(\mathbf{x},y)\sim\mathcal{D}} [\ell(y, h(\mathbf{x}))]$$

Let $\mathcal{H}$ be some class of functions from $\mathcal{X}$ to $\mathcal{Y}$. We define the loss of $\mathcal{H}$ with respect to the distribution $\mathcal{D}$ to be the loss of the best function in $\mathcal{H}$:

$$L_{\mathcal{D}}(\mathcal{H}) = \min_{h \in \mathcal{H}} L_{\mathcal{D}}(h)$$

So, $L_{\mathcal{D}}(\mathcal{H})$ measures whether $\mathcal{H}$ can approximate the distribution $\mathcal{D}$.

**The Class $\mathcal{F}$.** Our target functions will be parities on $k$ bits of the input. Let $A \subset [n]$ be some subset of size $|A| = k$, for some odd $k \geq 3$, and define $f_A$ to be the parity of the bits in $A$, namely $f_A(\mathbf{x}) = \text{sign}(\prod_{i \in A} x_i)$. For every subset $A \subset [n]$, we construct a distribution on the instances $\mathcal{X}$ that is easy to learn with neural-networks. Let $\mathcal{D}_A^{(1)}$ be the uniform distribution on $\mathcal{X}$, and let $\mathcal{D}_A^{(2)}$ be the distribution that is uniform on all the bits in $[n] \setminus A$, and the bits in $A$ are all 1 w.p. $\frac{1}{2}$ and $-1$ w.p. $\frac{1}{2}$. Let $\mathcal{D}_A$ be a distribution over $\mathcal{X} \times \mathcal{Y}$ where we samples $\mathbf{x} \sim \mathcal{D}_A^{(1)}$ w.p. $\frac{1}{2}$ and $\mathbf{x} \sim \mathcal{D}_A^{(2)}$ w.p. $\frac{1}{2}$, and set $y = f_A(\mathbf{x})$. This defines a family of distributions $\mathcal{F} = \{\mathcal{D}_A : A \subseteq [n], |A| = k\}$.

**Comment 1.** *Note that the above distributional assumptions are not typical. For example, it is more natural to assume that the underlying distribution is the uniform distribution. However, it is well-known that learning parities under the uniform distribution is computationally hard for statistical-query algorithms [17] and gradient-based algorithms [24]. Therefore, we must take an "easier" distribution to make the problem learnable using gradient-descent.*

**The training algorithm.** We train a neural-network with gradient-descent on the distribution $\mathcal{D}_A$. Let $g^{(t)} : \mathcal{X} \to \mathbb{R}$ be our neural-network at time $t$:

$$g^{(t)}(\mathbf{x}) = \sum_{i=1}^{2q} u_i^{(t)} \sigma \left( \left\langle \mathbf{w}_i^{(t)}, \mathbf{x} \right\rangle + b_i^{(t)} \right)$$

Where $u_i^{(t)}, b_i^{(t)} \in \mathbb{R}$, $\mathbf{w}_i^{(t)} \in \mathbb{R}^n$ and $\sigma$ denotes the ReLU6 activation $\sigma(x) = \min(\max(x, 0), 6)$. Define a regularization term $R(g^{(t)}) = \left\| \mathbf{u}^{(t)} \right\|^2 + \sum_{i=1}^{2q} \left\| \mathbf{w}_i^{(t)} \right\|^2$, and the hinge-loss function

$\ell(y, \hat{y}) = \max(1 - \hat{y}y, 0)$. Then, the loss on the distribution is $L_\mathcal{D}(g) = \mathbb{E}\left[\ell(y, g(\mathbf{x}))\right]$, and we perform the following updates:

$$\mathbf{w}^{(t)} = \mathbf{w}^{(t-1)} - \eta_t \frac{\partial}{\partial \mathbf{w}} \left( L_\mathcal{D}(g^{(t-1)}) + \lambda_t R(g^{(t-1)}) \right)$$

$$\mathbf{u}^{(t)} = \mathbf{u}^{(t-1)} - \eta_t \frac{\partial}{\partial \mathbf{u}} \left( L_\mathcal{D}(g^{(t-1)}) + \lambda_t R(g^{(t-1)}) \right)$$

for some choice of $\eta_1, \dots, \eta_T$ and $\lambda_1, \dots, \lambda_T$.

We assume the network is initialized with a symmetric initialization: for every $i \in [q]$ initialize $\mathbf{w}_i^{(0)} \sim U(\{-1, 0, 1\}^n)$ and then initialize $\mathbf{w}_{q+i}^{(0)} = -\mathbf{w}_i$, initialize $b_i^{(0)} = \frac{1}{8k}$ and $b_{q+i}^{(0)} = -\frac{1}{8k}$ and initialize $u_i^{(0)} \sim U([-\frac{n}{k}, \frac{n}{k}])$ and $u_{q+i}^{(0)} = -u_i^{(0)}$.

## 3 Main Result

Our main result shows a separation between neural-networks and *any* linear method (i.e., learning a linear classifier over some fixed embedding). This result is composed of two parts: first, we show that the family $\mathcal{F}$ *cannot* be approximated by any polynomial-size linear method. Second, we show that neural-networks *can* be trained to approximate the family $\mathcal{F}$ using gradient-descent.

The following theorem implies that the class $\mathcal{F}$ cannot be approximated by a linear classifiers on top of a fixed embedding, unless the embedding dimension or the norm of the weights is exponential:

**Theorem 2.** *Fix some $\Psi : \mathcal{X} \to [-1, 1]^N$, and define:*

$$\mathcal{H}_\Psi^B = \{\mathbf{x} \to \langle \Psi(\mathbf{x}), \mathbf{w} \rangle \ : \ \|\mathbf{w}\|_2 \le B\}$$

*Then, if $k \le \frac{n}{16}$, there exists some $\mathcal{D}_A \in \mathcal{F}$ such that:*

$$L_{\mathcal{D}_A}\left(\mathcal{H}_\Psi^B\right) \ge \frac{1}{2} - \frac{\sqrt{N}B}{2^k \sqrt{2}}$$

The following result shows that neural-networks can learn the family $\mathcal{F}$ with gradient-descent. That is, for every distribution $\mathcal{D}_A \in \mathcal{F}$, a large enough neural-network achieves a small error when trained with gradient-descent on the distribution $\mathcal{D}_A$. Together with theorem 2, it establishes an (exponential) separation between the class of distributions that can be learned with neural-networks, and the class of distributions that can be learned by linear methods.

**Theorem 3.** *Fix some $\mathcal{D}_A \in \mathcal{F}$. Assume we run gradient-descent for $T$ iterations, with $\eta_1 = 1, \lambda_1 = \frac{1}{2}$ and $\eta_t = \frac{k^2}{T\sqrt{q}}, \lambda_t \le \frac{k}{n}$ for every $t > 1$. Assume that $n \ge \Omega(1)$ and $7 \le k \le O(\sqrt[10]{n})$. Fix some $\delta > 0$, and assume that the number of neurons satisfies $q \ge \Omega(k^7 \log \frac{k}{\delta})$. Then, with probability at least $1 - \delta$ over the initialization, there exists $t \le T$ such that:*

$$\mathbb{P}\left[g^{(t)}(\mathbf{x}) \ne f_A(\mathbf{x})\right] \le L_{\mathcal{D}_A}\left(g^{(t)}\right) \le O\left(\frac{k^{10}}{q} + \frac{k^8}{\sqrt{q}} + \frac{qk}{\sqrt{n}} + \frac{k^2\sqrt{q}}{T}\right)$$

Observe that the above theorem immediately implies that neural networks of polynomial size, trained using gradient-descent for a polynomial number of steps, achieve small loss for every $\mathcal{D}_A \in \mathcal{F}$:

**Corollary 4.** *Fix $\epsilon, \delta \in (0, 1/2)$. For every $\mathcal{D}_A \in \mathcal{F}$, for large enough $n$, running gradient-descent with the assumptions in Theorem 3, achieves with probability at least $1 - \delta$ a loss of at most $\epsilon$ when training a network of size $q = poly(k, \epsilon^{-1}, \log 1/\delta)$ for $T = poly(k, \epsilon^{-1}, \log 1/\delta)$ iterations.*

*Proof.* Choose $q = \Theta(\epsilon^{-1}k^{16} \log(k/\delta))$, $T = \Theta(\epsilon^{-2}k^{10} \log(k/\delta))$ and $n \ge \Omega(\epsilon^{-4}k^{34} \log(k/\delta))$. □

## 4 Proof of Theorem 2

*Proof.* Let $\mathcal{L}_A(\mathbf{w}) := L_{\mathcal{D}_A^{(1)}}(\langle \Psi(\mathbf{x}), \mathbf{w} \rangle)$ and define the objective $G_A(\mathbf{w}) := \mathcal{L}_A(\mathbf{w}) + \frac{\lambda}{2} \|\mathbf{w}\|^2$. Observe that for every $i \in [N]$ we have:

$$\frac{\partial}{\partial w_i} G_A(0) = \mathop{\mathbb{E}}_{\mathbf{x} \sim \mathcal{D}} [f_A(\mathbf{x})\Psi_i(\mathbf{x})]$$

Since $\{f_A\}_{A \subseteq [n]}$ is a Fourier basis, we have:

$$\sum_{A \subseteq [n]} \mathop{\mathbb{E}}_{\mathbf{x} \sim \mathcal{D}} [f_A(\mathbf{x}) \Psi_i(\mathbf{x})]^2 = \|\Psi_i\|_2^2 \leq 1$$

And therefore:

$$\mathop{\mathbb{E}}_{A \subseteq [n], |A|=k} \left[ \|\nabla G_A(0)\|^2 \right] = \mathop{\mathbb{E}}_{A \subseteq [n], |A|=k} \left[ \sum_{i \in [N]} \left( \frac{\partial}{\partial w_i} G_A(0) \right)^2 \right]$$

$$= \mathop{\mathbb{E}}_{A \subseteq [n], |A|=k} \left[ \sum_{i \in [N]} \mathop{\mathbb{E}}_{\mathbf{x} \sim \mathcal{D}} [f_A(\mathbf{x}) \Psi_i(\mathbf{x})]^2 \right]$$

$$\leq \sum_{i \in [N]} \frac{1}{\binom{n}{k}} \sum_{A \subseteq [n]} \mathop{\mathbb{E}}_{\mathbf{x} \sim \mathcal{D}} [f_A(\mathbf{x}) \Psi_i(\mathbf{x})]^2 \leq \frac{N}{2^{4k}}$$

Where we use the fact that $\binom{n}{k} \geq (n/k)^k \geq 16^k$. Using Jensen inequality we get:

$$\mathop{\mathbb{E}}_{A} [\|\nabla G_A(0)\|]^2 \leq \mathop{\mathbb{E}}_{A} \left[ \|\nabla G_A(0)\|^2 \right] \leq \frac{N}{2^{4k}} \tag{1}$$

Note that $G_A$ is $\lambda$-strongly convex, and therefore, for every $\mathbf{w}, \mathbf{u}$ we have:

$$\langle \nabla G_A(\mathbf{w}) - \nabla G_A(\mathbf{u}), \mathbf{w} - \mathbf{u} \rangle \geq \lambda \|\mathbf{w} - \mathbf{u}\|^2$$

Let $\mathbf{w}_A^* := \arg\min_{\mathbf{w}} G_A(\mathbf{w})$, and so $\nabla G_A(\mathbf{w}_I^*) = 0$. Using the above we get:

$$\lambda \|\mathbf{w}_A^*\|^2 \leq \langle \nabla G_A(\mathbf{w}_A^*) - \nabla G_A(0), \mathbf{w}_A* \rangle \leq \|\nabla G_A(0)\| \|\mathbf{w}_A^*\| \Rightarrow \|\mathbf{w}_A^*\| \leq \frac{1}{\lambda} \|\nabla G_A(0)\|$$

Now, notice that $\mathcal{L}_A$ is $\sqrt{N}$-Lipschitz, since:

$$\|\nabla \mathcal{L}_A(\mathbf{w})\| = \|\nabla \mathbb{E} [\ell(y, \langle \Psi(\mathbf{x}), \mathbf{w} \rangle)]\| \leq \mathbb{E} [|\ell'| \|\psi(\mathbf{x})\|] \leq \sqrt{N}$$

Therefore, we get that:

$$1 - \mathcal{L}_A(\mathbf{w}_A^*) = \mathcal{L}_A(0) - \mathcal{L}_A(\mathbf{w}_A^*) \leq \sqrt{N} \|\mathbf{w}_A^*\| \leq \frac{\sqrt{N}}{\lambda} \|\nabla G_A(0)\| \tag{2}$$

Denote $\hat{\mathbf{w}}_A = \arg\min_{\|\mathbf{w}\| \leq B} \mathcal{L}_A(\mathbf{w})$, and by optimality of $\mathbf{w}_A^*$ we have:

$$\mathcal{L}_A(\mathbf{w}_A^*) \leq \mathcal{L}_A(\mathbf{w}_A^*) + \frac{\lambda}{2} \|\mathbf{w}_A^*\|^2 \leq \mathcal{L}_A(\hat{\mathbf{w}}_A) + \frac{\lambda}{2} \|\hat{\mathbf{w}}_A\|^2 \leq \mathcal{L}_A(\hat{\mathbf{w}}_A) + \frac{\lambda B^2}{2} \tag{3}$$

From (2) and (3) we get:

$$1 - \frac{\sqrt{N}}{\lambda} \|\nabla G_A(0)\| \leq \mathcal{L}_A(\mathbf{w}_A^*) \leq \mathcal{L}_A(\hat{\mathbf{w}}_A) + \frac{\lambda B^2}{2} \tag{4}$$

Taking an expectation and plugging in (1) we get:

$$\mathop{\mathbb{E}}_{A \subseteq [n], |A|=k} \left[ \min_{h \in \mathcal{H}_\Psi^B} L_{\mathcal{D}_A^{(1)}}(h) \right] = \mathbb{E} [\mathcal{L}_A(\hat{\mathbf{w}}_A)] \geq 1 - \frac{\sqrt{N}}{\lambda} \mathop{\mathbb{E}}_{A} [\|\nabla G_A(0)\|] - \frac{\lambda B^2}{2} \geq 1 - \frac{N}{\lambda 2^{2k}} - \frac{\lambda B^2}{2}$$

Since this is true for all $\lambda$, taking $\lambda = \frac{\sqrt{2N}}{2^k B}$ we get:

$$\mathop{\mathbb{E}}_{A \subseteq [n], |A|=k} \left[ \min_{h \in \mathcal{H}_\Psi^B} L_{\mathcal{D}_A^{(1)}}(h) \right] \geq 1 - \frac{\sqrt{2N} B}{2^k}$$

Therefore, there exists some $A \subseteq [n]$ with $|A| = k$ such that $\min_{h \in \mathcal{H}_\Psi^B} L_{\mathcal{D}_A^{(1)}}(h) \geq 1 - \frac{\sqrt{2N} B}{2^k}$.
Since $\mathcal{D}_A = \frac{1}{2} \mathcal{D}_A^{(1)} + \frac{1}{2} \mathcal{D}_A^{(2)}$ we get the required. $\qquad \square$

# 5 Proof of Theorem 3

We start by giving a rough sketch of the proof of Theorem 3. We divide the proof into two steps:

**First gradient step.** We show that after the first gradient step, there is a subset of "good" neurons in the first layer that approximately implement the function $\psi_j(\mathbf{x}) := \sigma(\tau_j \sum_{i \in A} x_i + b_j)$, for some $\tau_j$ and $b_j$. Indeed, observe that the correlation between every bit outside the parity and the label is zero, and so the gradient with respect to this bit becomes very small. However, for the bits in the parity, the correlation is large, and so the gradient is large as well.

**Convergence of online gradient-descent.** Notice that the parity can be implemented by a linear combination of the features $\psi_1(\mathbf{x}), \ldots, \psi_{q'}(\mathbf{x})$, when $\tau_1, \ldots, \tau_{q'}$ are distributed uniformly. Hence, from the previous argument, after the first gradient step there exists some choice of weights for the second layer that implements the parity (and hence, separates the distribution). Now, we show that for a sufficiently large network and sufficiently small learning rate, the weights of the first layer stay close to their value after the first iteration. Thus, a standard analysis of online gradient-descent shows that gradient-descent (on both layers) reaches a good solution.

In the rest of this section, we give a detailed proof, following the above sketch. For lack of space, some of the proofs for the technical lemmas appear in the appendix.

## 5.1 First Gradient Step

We want to show that for some "good" neurons, the weights $\mathbf{w}_i^{(1)}$ are close enough to $\tau_i \sum_{j \in A} e_j$ for some constant $\tau_i$ depending on $u_i^{(0)}$. We start by showing that the irrelevant coordinates ($j \notin A$) and the bias have very small gradient. To do this, we first analyze the gradient with respect to the uniform part of the distribution $\mathcal{D}_A^{(1)}$, and show that it is negligible, with high probability over the initialization of a neuron:

**Lemma 5.** *Fix $j \in [n] \setminus A$ and $b \in \mathbb{R}$ and let $\mathcal{D}$ be the uniform distribution. Let $f(\mathbf{x}) = \mathrm{sign}(\prod_{i \in A} x_i)$ be a parity. Then, for every $c > 0$, we have with probability at least $1 - \frac{1}{c}$ over the choice of $\mathbf{w}$:* $\left| \mathbb{E}_\mathbf{x} \, x_j f(\mathbf{x}) \cdot \sigma'(\mathbf{w}^\top \mathbf{x} + b > 0) \right| \leq c \sqrt{\frac{1}{\binom{n-1}{k}}}$. *A similar result holds for* $\left| \mathbb{E}_\mathbf{x} \, f(\mathbf{x}) \cdot \sigma'(\mathbf{w}^\top \mathbf{x} + b > 0) \right| \leq c \sqrt{\frac{1}{\binom{n-1}{k}}}$.

Using a union bound on the previous lemma, we get that the above result holds for all irrelevant coordinates (and the bias), with constant probability:

**Lemma 6.** *Let $k, n$ be odd numbers. Fix $b \in \mathbb{R}$ and let $\mathcal{D}$ be the uniform distribution. Let $f(\mathbf{x}) = \mathrm{sign}(\prod_{i \in A} x_i)$ be a parity. Then, for every $C > 0$, with probability at least $1 - \frac{1}{C}$ over the choice of $\mathbf{w}$:* $\forall j \notin A$, $\left| \mathbb{E}_\mathbf{x} \, x_j f(\mathbf{x}) \cdot \sigma'(\mathbf{w}^\top \mathbf{x} + b > 0) \right| \leq C(n-1) \sqrt{\frac{1}{\binom{n-1}{k}}}$ and $\left| \mathbb{E}_\mathbf{x} \, f(\mathbf{x}) \cdot \sigma'(\mathbf{w}^\top \mathbf{x} + b > 0) \right| \leq C(n-1) \sqrt{\frac{1}{\binom{n-1}{k}}}$.

*Proof.* of Lemma 6. Choose $c = C(n-1)$ and use union bound on the result of Lemma 5 over all choices of $j \notin A$ and the bias. $\square$

Now, we show that for neurons with $\sum_{j \in A} w_j = 0$, the gradient of the irrelevant coordinate and the bias is zero on the distribution $\mathcal{D}_A^{(2)}$ (the non-uniform part of the distribution $\mathcal{D}_A$):

**Lemma 7.** *Let $k, n$ be an odd numbers, let $f(\mathbf{x}) = \mathrm{sign}(\prod_{i \in A}^k x_i)$. Fix $\mathbf{w} \in \{1, 0, 1\}^n$ with $\sum_{i=1}^k w_i = 0$ and $b \in \mathbb{R}$. Then, on the distribution $\mathcal{D}_A^{(2)}$, we have:*

- $\mathbb{E} \left[ x_j f(\mathbf{x}) \cdot \sigma'(\mathbf{w}^\top \mathbf{x} + b > 0) \right] = 0$ *for all $j \notin A$*

- $\mathbb{E} \left[ f(\mathbf{x}) \cdot \sigma'(\mathbf{w}^\top \mathbf{x} + b > 0) \right] = 0$

Combining the above lemmas implies that for some "good" neurons, the gradient on the distribution $\mathcal{D}_A$ is negligible, for the irrelevant coordinates and the bias. Now, it is left to show that for the

coordinates of the parity ($j \in A$), the gradient on the distribution $\mathcal{D}_A$ is large. To do this, we show that the gradient on the relevant coordinates is almost independent from the gradient of the activation function. Since the gradient with respect to the hinge-loss at the initialization is simply the correlation, this is sufficient to show that the gradient of the relevant coordinates is large.

**Lemma 8.** *Observe the distribution $\mathcal{D}_A$. Let $h : \mathcal{X} \to \{\pm 1\}$ some function supported on $A$. Let $\mathbf{w} \in \{-1, 0, 1\}^n$ be some vector, and $b \in \mathbb{R}$. Denote $J = \{j \in [n] \setminus A \;:\; w_j \neq 0\}$ and denote $\varphi(\mathbf{w}, b) = \mathbb{P}\left[\frac{k}{\sqrt{n}} < \sum_{j \in J} w_j x_j + b < 6 - \frac{k}{\sqrt{n}}\right]$. Then there exists a universal constants $C$ s.t.:*

$$\left|\mathbb{E}\left[h(\mathbf{x}) \cdot \sigma'(\mathbf{w}^\top \mathbf{x} + b)\right] - \mathbb{E}\left[h(\mathbf{x})\right] \varphi(\mathbf{w}, b)\right| \leq \frac{Ck}{\sqrt{|J|}}$$

From all the above, we get that with non-negligible probability, the weights of a given neuron are approximately $\alpha_i \sum_{j \in A} e_j$, for some choice of $\alpha_i$ depending on $u_i^{(0)}$:

**Lemma 9.** *Assume $n \geq 9 \log 7$ and $7 \leq k \leq \frac{1}{2\sqrt{2}} \sqrt[4]{n}$. Fix some $i \in [q]$. Then, with probability at least $\frac{1}{14\sqrt{k}}$ over the choice of $\mathbf{w}_i^{(0)}$, we have that: $\max_{j \in A} \left|w_{i,j}^{(1)} - \frac{\alpha_i}{\sqrt{n}} u_i^{(0)}\right| \leq C_1$, $\max_{j \notin A} \left|w_{i,j}^{(1)}\right| \leq \frac{C_2}{n-1}$ and $\left|b^{(1)} - b^{(0)}\right| \leq \frac{C_3}{\sqrt{n}}$ for some universal constants $C_1, C_2, C_3$, and some $\alpha_i \in \left[\frac{1}{4}, 1\right]$ depending on $\mathbf{w}_i^{(0)}$.*

Finally, we show that the features implemented by the "good" neurons can express the parity function, using a linear separator with low norm. In the next two lemmas we show explicitly what are the features that the "good" neurons approximate:

**Lemma 10.** *Let $b = \frac{1}{8k}$, $\alpha \in \left[\frac{1}{4}, 1\right]$ and $r \in \{-k, -k+2, \ldots, k-2, k\}$. Denote $\phi_r(z) = \sigma(-\operatorname{sign}(r)z + |r|)$. Let $u \sim U\left(\left[-\frac{n}{k}, \frac{n}{k}\right]\right)$. Then, for every $\epsilon \leq kr$, with probability at least $\frac{\epsilon}{8k^2}$ over the choice of $u$ we have $\left|\frac{|r|}{b}\sigma\left(\frac{\alpha}{n} uz + b\right) - \phi_r(z)\right| \leq \epsilon$, for every $z \in [-k, k]$.*

**Lemma 11.** *Fix $\epsilon > 0$ and assume that $n \geq 9 \log 7$ and $7 \leq k \leq c\sqrt[4]{\epsilon}\sqrt[8]{n}$, for some universal constant $c$. Fix $r \in \{-k, -k+2, \ldots, k-2, k\}$, $\epsilon \leq kr$ and define $\psi_r(\mathbf{x}) = \sigma\left(-\operatorname{sign}(r)\sqrt{n}\sum_{j \in A} x_j + |r|\right)$. Fix some $i \in [q]$. Then, with probability at least $\frac{\epsilon}{112k^{2.5}}$ over the choice of $\mathbf{w}_i^{(0)}, u_i^{(0)}$ then for $\widehat{\psi}_i(\mathbf{x}) = \frac{|r|}{b_i^{(0)}}\sigma\left(\left\langle \mathbf{w}_i^{(1)}, \mathbf{x}\right\rangle + b_i^{(1)}\right)$ we have $\left|\widehat{\psi}_r(\mathbf{x}) - \psi_r(\mathbf{x})\right| \leq 2\epsilon$ for all $\mathbf{x} \in \mathcal{X}$.*

Using the above, we show that there exists a choice for the weights for the second layer that implement the parity, with high probability over the initialization:

**Lemma 12.** *Assume that $n \geq 9 \log 7$ and $7 \leq k \leq c\sqrt[10]{n}$, for some universal constant $c$. Fix some $\delta > 0$, and assume that the number of neurons satisfies $q \geq Ck^7 \log\left(\frac{k+1}{\delta}\right)$. Then, with probability at least $1 - \delta$ over the choice of the weights, there exists $\mathbf{u}^* \in \mathbb{R}^{2q}$ such that $g^*(\mathbf{x}) = \sum_{i=1}^{2q} u_i^* \sigma\left(\left\langle \mathbf{w}_i^{(1)}, \mathbf{x}\right\rangle + b_i^{(1)}\right)$ satisfies $g^*(\mathbf{x})f_A(\mathbf{x}) \geq 1$ for all $\mathbf{x} \in \mathcal{X}$. Furthermore, we have $\|\mathbf{u}^*\|_2 \leq B\frac{k^5}{\sqrt{q}}$, $\|u^*\|_0 = \tilde{B}\frac{q}{k^{2.5}}$, and for every $i \in [2q]$ with $u_i^* \neq 0$ we have $\sigma\left(\left\langle \mathbf{w}_i^{(1)}, \mathbf{x}\right\rangle + b_i^{(1)}\right) \leq 1$, for some universal constants $B, \tilde{B}$.*

This concludes the analysis of the first gradient step.

## 5.2 Convergence of Gradient-Descent

Our main result in this part relies on the standard analysis of online gradient-descent. Specifically, this analysis shows that performing gradient-descent on a sequence of convex functions reaches a set of parameters that competes with the optimum (in hindsight). We give this result in general, when we optimize the functions $f_1, \ldots, f_t$ with respect to the parameter $\theta$:

**Theorem 13.** *(Online Gradient Descent) Fix some $\eta$, and let $f_1, \ldots, f_T$ be some sequence of convex functions. Fix some $\theta_1$, and assume we update $\theta_{t+1} = \theta_t - \eta \nabla f_t(\theta_t)$. Then for every $\theta^*$ the following holds:*

$$\frac{1}{T}\sum_{t=1}^{T} f_t(\theta_t) \leq \frac{1}{T}\sum_{t=1}^{T} f_t(\theta^*) + \frac{1}{2\eta T}\|\theta^*\|^2 + \|\theta_1\|\frac{1}{T}\sum_{t=1}^{T}\|\nabla f_t(\theta_t)\| + \eta\frac{1}{T}\sum_{t=1}^{T}\|\nabla f_t(\theta_t)\|^2$$

Note that in the previous part we showed that the value of the weights of the first layer is "good" with high probability. In other words, optimizing only the second layer after the first gradient step is sufficient to achieve a good solution. However, since we optimize both layers with gradient-descent, we need to show that the weights of the first layer stay close to their value after the first initialization. We start by bounding the weights of the second layer after the first iteration:

**Lemma 14.** *Assume $\eta_1 = 1$ and $\lambda_1 = \frac{1}{2}$. Then for every $i \in [q]$ we have $\left| u_i^{(1)} \right| \leq \frac{k}{\sqrt{n}}$.*

Using this, we can bound how much the first layer changes after at every gradient-step:

**Lemma 15.** *Assume that $\eta_1 = 1, \lambda_1 = \frac{1}{2}$ and $\eta_t = \eta, \lambda_t = \lambda$ for every $t > 1$, for some fixed value $\eta, \lambda \in [0, \frac{1}{2}]$. For every $t$ and every $i \in [2q]$ we have $\left| u_i^{(t)} \right| \leq +6\eta t + \frac{k}{\sqrt{n}}, \left\| \mathbf{w}_i^{(t)} - \mathbf{w}_i^{(1)} \right\| \leq 6\eta^2 t^2 + \eta t \frac{k}{\sqrt{n}} 2\eta t \lambda \frac{n}{k} t$ and $\left| b_i^{(t)} - b_i^{(1)} \right| \leq 6\eta^2 t^2 + \eta t \frac{k}{\sqrt{n}}$.*

Using the above we bound the difference in the loss between optimizing the first layer and keeping it fixed, for every choice of $\mathbf{u}^*$ for the second layer:

**Lemma 16.** *Fix some vector $\mathbf{u}^* \in \mathbb{R}^{2q}$, and let $g_{\mathbf{u}^*}^{(t)}(\mathbf{x}) = \sum_{i=1}^{2q} u_i^* \sigma \left( \left\langle \mathbf{w}_i^{(t)}, \mathbf{x} \right\rangle + b_i^{(t)} \right)$. Then we have: $\left| \ell(g_{\mathbf{u}^*}^{(t)}(\mathbf{x}), y) - \ell(g_{\mathbf{u}^*}^{(1)}(\mathbf{x}), y) \right| \leq 2 \left\| \mathbf{u}^* \right\|_2 \sqrt{\left\| \mathbf{u}^* \right\|_0} \left( 6\eta^2 t^2 + \eta t \frac{k}{\sqrt{n}} + \eta t \lambda \frac{n}{k} \right)$.*

Finally, using all the above we can prove our main theorem:

*Proof.* of Theorem 3. Let $\mathbf{u}^* \in \mathbb{R}^{2q}$ be the separator from Lemma 12, and we have $\left\| \mathbf{u}^* \right\|_2 \leq B \frac{k^5}{\sqrt{q}}$ and $\|u^*\|_0 = \tilde{B} \frac{q}{k^{2.5}}$. Denote $\tilde{L}_{\mathcal{D}}(g^{(t)}) = \mathbb{E}\left[ \ell(g(\mathbf{x}), y) \right] + \lambda_t \left\| \mathbf{u}^{(t)} \right\|^2$, and notice that the gradient of $\tilde{L}_{\mathcal{D}}$ with respect to $\mathbf{u}$ is the same as the gradient of the original objective. From Lemma 14, we have $\left\| \mathbf{u}^{(1)} \right\| \leq \frac{\sqrt{2q}k}{\sqrt{n}}$. Since $\tilde{L}_{\mathcal{D}}$ is convex with respect to $\mathbf{u}$, from Theorem 13 we have:

$$\frac{1}{T} \sum_{t=2}^{T+1} \tilde{L}_{\mathcal{D}}(g^{(t)})$$

$$\leq \frac{1}{T} \sum_{t=2}^{T+1} \tilde{L}_{\mathcal{D}}(g_{\mathbf{u}^*}^{(t)}) + \frac{\|\mathbf{u}^*\|^2}{2\eta T} + \frac{\|\mathbf{u}^{(1)}\|}{T} \sum_{t=2}^{T+1} \left\| \frac{\partial}{\partial \mathbf{u}} \tilde{L}_{\mathcal{D}}(g^{(t)}) \right\| + \frac{\eta}{T} \sum_{t=2}^{T+1} \left\| \frac{\partial}{\partial \mathbf{u}} \tilde{L}_{\mathcal{D}}(g^{(t)}) \right\|^2$$

$$\leq \frac{1}{T} \sum_{t=2}^{T+1} \tilde{L}_{\mathcal{D}}(g_{\mathbf{u}^*}^{(t)}) + \frac{B^2 k^{10}}{2\eta Tq} + \frac{6\sqrt{2}qk}{\sqrt{n}} + 36\eta q$$

Using Lemma 16 we get that for every $t$ we have:

$$\left| \tilde{L}_{\mathcal{D}}(g_{\mathbf{u}^*}^{(t)}) - \tilde{L}_{\mathcal{D}}(g_{\mathbf{u}^*}^{(1)}) \right| \leq 2 \left\| \mathbf{u}^* \right\|_2 \sqrt{\left\| \mathbf{u}^* \right\|_0} \left( 6\eta^2 t^2 + \eta t \frac{k}{\sqrt{n}} + \eta t \lambda \frac{n}{k} \right)$$

$$\leq B' k^4 \left( 6\eta^2 T^2 + \eta T \right)$$

Therefore we get:

$$\frac{1}{T} \sum_{t=2}^{T+1} \tilde{L}_{\mathcal{D}}(g^{(t)}) \leq \tilde{L}_{\mathcal{D}}(g_{\mathbf{u}^*}^{(1)}) + B' k^4 \left( 6\eta^2 T^2 + \eta T \right) + \frac{B^2 k^{10}}{2\eta Tq} + \frac{6\sqrt{2}qk}{\sqrt{n}} + 36\eta q$$

Now, take $\eta = \frac{k^2}{T\sqrt{q}}$. Since $\mathbf{u}^*$ separates the distribution $\mathcal{D}$ with margin 1, when taking the weights after the first iteration, we have $\tilde{L}_{\mathcal{D}}(g_{\mathbf{u}^*}^{(1)}) \leq \frac{1}{2} \left\| \mathbf{u}^* \right\|^2 = B^2 \frac{k^{10}}{2q}$. Therefore:

$$\frac{1}{T} \sum_{t=2}^{T+1} \tilde{L}_{\mathcal{D}}(g^{(t)}) \leq B^2 \frac{k^{10}}{2q} + B' \frac{6k^8}{q} + B' \frac{k^6}{\sqrt{q}} + \frac{B^2 k^8}{2\sqrt{q}} + \frac{6\sqrt{2}qk}{\sqrt{n}} + \frac{36k^2\sqrt{q}}{T}$$

From this, there exists some $2 \leq t \leq T+1$ such that:

$$\tilde{L}_{\mathcal{D}}(g^{(t)}) \leq B^2 \frac{k^{10}}{2q} + B' \frac{6k^8}{q} + B' \frac{k^6}{\sqrt{q}} + \frac{B^2 k^8}{2\sqrt{q}} + \frac{6\sqrt{2}qk}{\sqrt{n}} + \frac{36k^2\sqrt{q}}{T}$$

And since the hinge-loss upper bounds the zero-one loss, we get the required. ☐

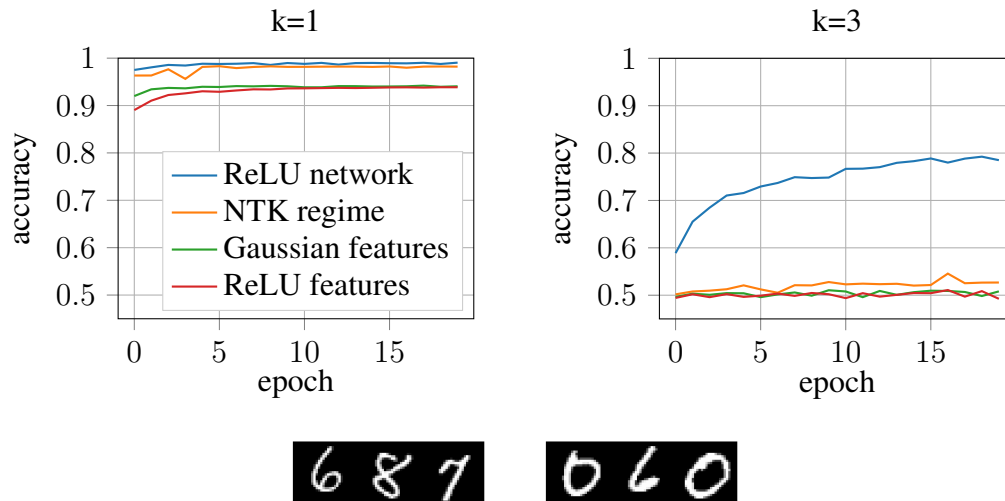

Figure 1: MNIST-parity experiment. Top left: test performance on parity of a single MNIST image. Top right: test performance on parity of three MNIST images. Bottom: examples for the MNIST-parity experiment, the model has to predict the parity of the digits sum.

## 6 Experiment

In section 3 we showed a family of distributions $\mathcal{F}$ that separates linear classes from neural-networks. To validate that our theoretical results apply to a more realistic setting, we perform an experiment that imitates the parity problem using the MNIST dataset. We observe the following simple task: given a strip with $k$ digits chosen uniformly at random from the MNIST dataset, determine whether the sum of the digits is even or odd. We compare the performance of a ReLU network with one hidden-layer, against various linear models.

In the case where $k = 1$, the MNIST-parity task is just a simplified version of the standard MNIST classification task, where instead of 10 classes there are only 2 classes of numbers. In this case, we observe that both the neural-network model and the linear models obtain similar performance, with only slight advantage to the neural-network model. However, when $k = 3$, the task becomes much harder: it is not enough to merely memorize the digits and assign them to classes, as the model needs to compute the parity of their sum. In this case, we observe a dramatic gap between the performance of the ReLU network and the performance of the linear models. While the ReLU network achieves performance of almost $80\%$ accuracy, the linear models barely perform better than a chance. The results of the experiment are shown in Figure 1.

### 6.1 Experiment Details

In the MNIST-parity experiment we train a neural-network model, as well as various linear models, to predict the parity of the sum of digits in the strip. Our neural-network architecture is a one-hidden layer network with ReLU activation and 512 neurons in the hidden layer. We compare this network to a network of a similar architecture, except that we force the network to stay in the regime of the linear approximation induced by the network's gradient - i.e., the neural-tangent-kernel (NTK) regime [15]. To do this, we use an architecture that decouples the gating from the linearity of the ReLU funcion, and keeps the gates fixed throughout the training process (as suggested in [12]). Namely, we use the fact that: $\text{ReLU}(\langle \mathbf{w}, \mathbf{x} \rangle) = \langle \mathbf{w}, \mathbf{x} \rangle \cdot \mathbf{1}\{\langle \mathbf{w}, \mathbf{x} \rangle\}$, and by decoupling the first and second term during the optimization, we force the network to stay in the NTK regime. We compare these architectures to standard random-features models, where we randomly initialize the first layer, but train only the second layer. Such models are known to be an efficient method for approximating kernel-SVM (see [22]). We use both ReLU random-features (standard random initialization with ReLU activation), and Gaussian random features, which approximate the RBF kernel. Both models have 512 features in the first layer. All models are trained with AdaDelta optimizer, for 20 epochs, with batch size 128.

# 7 Discussion and Future Work

In this work we showed exponential separation between learning neural networks with gradient-descent and learning linear models - i.e., learning linear separators over fixed representation of the data. This shows that learning neural networks is a strictly stronger learning model than any linear model, including linear classifiers, kernel methods and random features. In other words, neural networks are not just "glorified" kernel methods, as might be implied from previous works in the field. This demonstrates that our current understanding of neural networks learning is very limited, as only a few works so far have given positive results beyond the linear case.

There are various open questions which we leave for future work. The first immediate research direction is to find other distribution families that are learnable with neural networks via gradient-descent, but not using linear models. Another interesting question is finding distribution families with separation between deep and shallow networks. Specifically, finding a family of distributions that are learnable with gradient-descent using depth-three networks, but cannot be learned using depth-two networks. Finally, we believe that understanding the behavior of neural networks trained on specific "non-linear" distribution families will allow us to induce specific properties of the distributions that make them learnable using neural networks. Characterizing such distributional properties is another promising direction for future research.

## Broader Impact

As the primary focus of this paper is on theoretical results and theoretical analysis, a Broader Impact discussion is not applicable.

## Acknowledgments and Disclosure of Funding

This research is partially supported by ISF grant 2258/19.

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
