[Supplementary Material]

# A Additional Proof Details

*Proof.* of Lemma 5. Fix some $\mathbf{w}$. Denote $h(\mathbf{x}) = x_j \cdot \sigma'(\mathbf{w}^\top \mathbf{x} + b > 0)$. Let $A' \subseteq [n]$ be some subset with $|A'| = k$ and $j \notin A'$.

$$\mathbb{E}\, x_j f_{A'}(\mathbf{x}) \cdot \sigma'(\mathbf{w}^\top \mathbf{x} + b > 0) = \hat{h}(A')$$

Now, we have

$$\mathbb{E}_{A'} \left|\hat{h}(A')\right|^2 = \frac{1}{\binom{n-1}{k}} \sum_{A' \in \binom{[n-1]}{k}} \left|\hat{h}(A')\right|^2 \leq \frac{\|h\|_2^2}{\binom{n-1}{k}} \leq \frac{1}{\binom{n-1}{k}}$$

Finally,

$$\mathbb{E}_{A'} \left|\hat{h}(A')\right| \leq \sqrt{\mathbb{E}_{A'} \left|\hat{h}(A)\right|^2} \leq \sqrt{\frac{1}{\binom{n-1}{k}}}$$

Since the above holds for all $\mathbf{w}$, we get that:

$$\mathbb{E}_{A'} \mathbb{E}_{\mathbf{w}} \left|\hat{h}(A')\right| = \mathbb{E}_{\mathbf{w}} \mathbb{E}_{A'} \left|\hat{h}(A')\right| \leq \sqrt{\frac{1}{\binom{n-1}{k}}}$$

Fix some $A' \subseteq [n]$ (with $|A'| = k$ and $j \notin A'$), and observe that, from symmetry to permutations of the uniform distribution, we have:

$$\mathbb{E}_{\mathbf{w}} \left|\hat{h}(A')\right| = \mathbb{E}_{\mathbf{w}} \left|\mathbb{E}_{\mathbf{x}} x_j f_{A'}(\mathbf{x}) \cdot \sigma'(\mathbf{w}^\top \mathbf{x} + b > 0)\right|$$
$$= \mathbb{E}_{\mathbf{w}} \left|\mathbb{E}_{\mathbf{x}} x_j f_A(\mathbf{x}) \cdot \sigma'(\mathbf{w}^\top \mathbf{x} + b > 0)\right| = \mathbb{E}_{\mathbf{w}} \left|\hat{h}(A)\right|$$

And therefore, we get that: $\mathbb{E}_{\mathbf{w}} \left|\hat{h}(A)\right| = \mathbb{E}_{A'} \mathbb{E}_{\mathbf{w}} \left|\hat{h}(A')\right| \leq \sqrt{\frac{1}{\binom{n-1}{k}}}$. Now, using Markov's inequality achieves the required. A similar calculation is valid for $h(\mathbf{x}) = \sigma'(\mathbf{w}^\top \mathbf{x} + b > 0)$. $\square$

*Proof.* of Lemma 7. W.l.o.g., assume $A = [k]$ and $j = k+1$. We will show that the conclusion of the lemma is true even if we condition of the value of $x_{k+1}, \ldots, x_n$. Indeed, in that case the conditional expectation of $x_j f(\mathbf{x}) \cdot \sigma'(\mathbf{w}^\top \mathbf{x} + b > 0)$ is

$$\frac{1}{2} x_{k+1} f(1, \ldots, 1, x_{k+1} \ldots, x_k) \cdot \sigma' \left( \sum_{i=1}^{k} w_i + \sum_{i=k+1}^{n} w_i x_i + b > 0 \right)$$

$$+ \frac{1}{2} x_{k+1} f(-1, \ldots, -1, x_{k+1} \ldots, x_k) \cdot \sigma' \left( \sum_{i=1}^{k} -w_i + \sum_{i=1}^{n} w_i x_i + b > 0 \right)$$

$$= \frac{1}{2} x_{k+1} \cdot \sigma' \left( \sum_{i=k+1}^{n} w_i x_i + b > 0 \right)$$

$$- \frac{1}{2} x_{k+1} \cdot \sigma' \left( \sum_{i=k+1}^{n} w_i x_i + b > 0 \right)$$

$$= 0$$

Similarly, the conditional expectation of $f(\mathbf{x}) \cdot \sigma'(\mathbf{w}^\top \mathbf{x} + b > 0)$ is

$$
\frac{1}{2} f(1, \ldots, 1, x_{k+1} \ldots, x_k) \cdot \sigma'\left(\sum_{i=1}^{k} w_i + \sum_{i=k+1}^{n} w_i x_i + b > 0\right)
$$

$$
+ \frac{1}{2} f(-1, \ldots, -1, x_{k+1} \ldots, x_k) \cdot \sigma'\left(\sum_{i=1}^{k} -w_i + \sum_{i=1}^{n} w_i x_i + b > 0\right)
$$

$$
= \quad \frac{1}{2} \cdot \sigma'\left(\sum_{i=k+1}^{n} w_i x_i + b > 0\right)
$$

$$
- \frac{1}{2} \cdot \sigma'\left(\sum_{i=k+1}^{n} w_i x_i + b > 0\right)
$$

$$
= \quad 0
$$

$\square$

*Proof.* of Lemma 8. Fix some $y \in \{\pm 1\}$. Denote $\hat{S}$ to be the random variable $\hat{S} := \sum_{j \notin A} w_j x_j = \sum_{j \in J} w_j x_j$. Notice that for every $y \in \{\pm 1\}$, the following holds:

$$
\mathbb{P}\left[h(\mathbf{x}) = y \wedge \sigma'(\mathbf{w}^\top \mathbf{x} + b) = 1\right] \leq \mathbb{P}\left[h(\mathbf{x}) = y \wedge \hat{S} + b \in (\frac{k}{\sqrt{n}}, 6 - \frac{k}{\sqrt{n}})\right]
$$

$$
+ \mathbb{P}\left[h(\mathbf{x}) = y \wedge \hat{S} + b \in (-\frac{k}{\sqrt{n}}, \frac{k}{\sqrt{n}}] \cup [6 - \frac{k}{\sqrt{n}}, 6)\right]
$$

$$
= \mathbb{P}\left[h(\mathbf{x}) = y\right] \mathbb{P}\left[\hat{S} + b \in (\frac{k}{\sqrt{n}}, 6 - \frac{k}{\sqrt{n}})\right]
$$

$$
+ \mathbb{P}\left[h(\mathbf{x}) = y \wedge \hat{S} + b \in (-\frac{k}{\sqrt{n}}, \frac{k}{\sqrt{n}}] \cup [6 - \frac{k}{\sqrt{n}}, 6)\right]
$$

Where we use the fact that $h(\mathbf{x})$ is independent from every $x_j$ with $j \notin A$. Since $\{\sqrt{n} x_j\}_{j \in J}$ are Rademacher random variables, from Littlewood-Offord there exists a universal constant $B$ such that $\mathbb{P}\left[\hat{S} \in I\right] \leq \frac{B}{\sqrt{|J|}}$, for every open interval $I$ of length $\frac{1}{\sqrt{n}}$. Using the union bound we get that $\mathbb{P}\left[\hat{S} + b \in (-\frac{k}{\sqrt{n}}, \frac{k}{\sqrt{n}}] \cup [6 - \frac{k}{\sqrt{n}}, 6)\right] \leq \frac{3k+2}{\sqrt{|J|}}$. Therefore, we get the following:

$$
\left|\mathbb{P}\left[h(\mathbf{x}) = y \wedge \sigma'(\mathbf{w}^\top \mathbf{x} + b) = 1\right] - \mathbb{P}\left[h(\mathbf{x}) = y\right] \mathbb{P}\left[\hat{S} + b \in (\frac{k}{\sqrt{n}}, 6 - \frac{k}{\sqrt{n}})\right]\right|
$$

$$
\leq \mathbb{P}\left[h(\mathbf{x}) = y \wedge \hat{S} + b \in (-\frac{k}{\sqrt{n}}, \frac{k}{\sqrt{n}}] \cup [6 - \frac{k}{\sqrt{n}}, 6)\right]
$$

$$
= \mathbb{P}\left[h(\mathbf{x}) = y\right] \mathbb{P}\left[\hat{S} + b \in (-\frac{k}{\sqrt{n}}, \frac{k}{\sqrt{n}}] \cup [6 - \frac{k}{\sqrt{n}}, 6)\right]
$$

$$
\leq \mathbb{P}\left[h(\mathbf{x}) = y\right] \frac{(3k+1)B}{\sqrt{|J|}}
$$

Since the above is true for every $y \in \{\pm 1\}$, we get that:

$$\left| \mathbb{E}\left[h(\mathbf{x}) \cdot \sigma'(\mathbf{w}^\top \mathbf{x} + b)\right] - \mathbb{E}\left[h(\mathbf{x})\right] \mathbb{P}\left[\hat{S} + b \in (\frac{k}{\sqrt{n}}, 6 - \frac{k}{\sqrt{n}})\right] \right|$$

$$= \left| \sum_{y \in \{\pm 1\}} y \mathbb{P}\left[h(\mathbf{x}) = y \wedge \sigma'(\mathbf{w}^\top \mathbf{x} + b) = 1\right] - \sum_{y \in \{\pm 1\}} y \mathbb{P}\left[h(\mathbf{x}) = y\right] \mathbb{P}\left[\hat{S} + b \in (\frac{k}{\sqrt{n}}, 6 - \frac{k}{\sqrt{n}})\right] \right|$$

$$\leq \sum_{y \in \{\pm 1\}} \left| \mathbb{P}\left[h(\mathbf{x}) = y \wedge \sigma'(\mathbf{w}^\top \mathbf{x} + b) = 1\right] - \mathbb{P}\left[h(\mathbf{x}) = y\right] \mathbb{P}\left[\hat{S} + b \in (\frac{k}{\sqrt{n}}, 6 - \frac{k}{\sqrt{n}})\right] \right|$$

$$\leq \frac{(3k+1)B}{\sqrt{|J|}} \sum_{y \in \{\pm 1\}} \mathbb{P}\left[h(\mathbf{x}) = y\right] = \frac{(3k+1)B}{\sqrt{|J|}}$$

And this gives the required. $\qquad\square$

*Proof.* of Lemma 9. Denote $\mathbf{w} := \mathbf{w}_i^{(0)}$, $b := b_i^{(0)}$. We show that with probability at least $\frac{1}{14\sqrt{k}}$ over the choice of $\mathbf{w}_i^{(0)}$ we have:

1. $\left| \mathbb{E}_\mathbf{x} x_j f(\mathbf{x}) \cdot \sigma'(\mathbf{w}^\top \mathbf{x} + b) \right| \leq 14\sqrt{k}(n-1)\sqrt{\frac{1}{\binom{n-1}{k}}}$

   $\left| \mathbb{E}_\mathbf{x} f(\mathbf{x}) \cdot \sigma'(\mathbf{w}^\top \mathbf{x} + b) \right| \leq 14\sqrt{k}(n-1)\sqrt{\frac{1}{\binom{n-1}{k}}}$

2. $\sum_{j \in A} w_j = 0$

3. $|J| := |\{j \in [n] \setminus A : w_j \neq 0\}| \geq \frac{n-k}{3}$

We start by calculating the probability to get each of the above separately:

1. From Lemma 6, this holds with probability at least $1 - \frac{1}{14\sqrt{k}}$.

2. Denote $A_0 = \{j \in A \mid w_j = 0\}$. Now, to calculate the probability that 2 holds, we start by noting that it can hold only when $|A_0|$ is odd (since $k$ is odd). Now, note that $\mathbb{P}\left[w_j = 0\right] = \frac{1}{3}$ independently for every coordinate. Therefore, we have the following:

$$((1 - \frac{1}{3}) + \frac{1}{3})^k = \mathbb{P}\left[|A_0| \text{ is even}\right] + \mathbb{P}\left[|A| \text{ is odd}\right]$$

$$((1 - \frac{1}{3}) - \frac{1}{3})^k = \mathbb{P}\left[|A_0| \text{ is even}\right] - \mathbb{P}\left[|A| \text{ is odd}\right]$$

$$\Rightarrow \mathbb{P}\left[|A_0| \text{ is odd}\right] = \frac{1}{2} - \frac{1}{2}(\frac{1}{3})^k \geq \frac{1}{3}$$

Now, conditioning on the event that $|A_0|$ is odd, we have:

$$\mathbb{P}\left[\sum_{j \in A} w_j = 0\right] = \frac{1}{2^{k-|A_0|}}\binom{k - |A_0|}{\frac{1}{2}(k - |A_0|)} \geq \frac{1}{2\sqrt{k - |A_0|}} \geq \frac{1}{2\sqrt{k}}$$

All in all, we get that 2 holds with probability at least $\frac{1}{6\sqrt{k}}$.

3. Denote $X_j = \mathbf{1}\{w_j \neq 0\}$, and note that we have $\mathbb{E}\left[\sum_{j \notin A} X_j\right] = \frac{2(n-k)}{3}$. Then, from Hoeffding's inequality we get that $\mathbb{P}\left[|J| \leq \frac{n-k}{3}\right] \leq \exp(-\frac{2}{9}(n-k)) \leq \frac{1}{7}$, since we assume $n - k \geq \frac{9}{2}\log 7$.

To calculate the probability that both 1,2 and 3 hold, note that 2 and 3 are independent, and therefore the probability that both of them hold is at least $\frac{1}{7\sqrt{k}}$. Using the union bound we get that the probability that all 1-3 hold is at least $\frac{1}{14\sqrt{k}}$.

Now, we assume that the above hold. In this case we have:

$$\left| b_i^{(1)} - b_i^{(0)} \right| = \left| \eta_1 \frac{\partial}{\partial b_i} L_{\mathcal{D}}(g^{(0)}) \right|$$

$$= \left| \mathbb{E} \left[ \ell'(f_A(\mathbf{x}), g^{(0)}(\mathbf{x})) \frac{\partial}{\partial b_i} g^{(0)}(\mathbf{x}) \right] \right|$$

$$= \left| u_i^{(0)} \right| \left| \frac{1}{2} \mathbb{E}_{\mathcal{D}_A^{(1)}} f_A(\mathbf{x}) \cdot \sigma'(\mathbf{w}^\top \mathbf{x} + b) - \frac{1}{2} \mathbb{E}_{\mathcal{D}_A^{(2)}} f_A(\mathbf{x}) \cdot \sigma'(\mathbf{w}^\top \mathbf{x} + b) \right|$$

$$= \frac{n}{2k} \left| \mathbb{E}_{\mathcal{D}_A^{(1)}} f(\mathbf{x}) \cdot \sigma'(\mathbf{w}^\top \mathbf{x} + b) \right|$$

$$\le 7 \frac{n(n-1)}{\sqrt{k}} \sqrt{\frac{1}{\binom{n-1}{k}}} \le 7(n-1)^2 \sqrt{\frac{1}{\binom{n-1}{5}}} \le 7(n-1)^2 \sqrt{\left( \frac{5}{n-1} \right)^5} \le \frac{\sqrt{2} \cdot 7 \cdot 5^{2.5}}{\sqrt{n}}$$

Where we use the result of Lemma 7 and the above conditions. Now, for all $j \in [n]$ we have:

$$w_{i,j}^{(1)} = w_{i,j}^{(0)} - \eta_1 \left( \frac{\partial}{\partial w_{i,j}} L_{\mathcal{D}}(g^{(0)}) + \lambda_1 R(g^{(0)}) \right)$$

$$= w_i^{(0)} - \mathbb{E} \left[ \ell'(f_A(\mathbf{x}), g^{(0)}(\mathbf{x})) \frac{\partial}{\partial w_{i,j}} g^{(0)}(\mathbf{x}) \right] - \frac{1}{2} \frac{\partial}{\partial w_{i,j}^{(0)}} R(g^{(0)})$$

$$= -u_i^{(0)} \mathbb{E} \left[ x_j f_A(\mathbf{x}) \sigma'(\mathbf{w}^\top \mathbf{x} + b) \right]$$

So, denote $h(\mathbf{x}) = \sqrt{n} x_j f_A(\mathbf{x})$ and note that for every $j \in A$ we get $h(\mathbf{x}) \equiv 1$. So, from Lemma 8 we get that for every $j \in A$ we have:

$$\left| w_{i,j}^{(1)} - \varphi(\mathbf{w}, b) \frac{u_i^{(0)}}{\sqrt{n}} \right| = \left| \frac{u_i^{(0)}}{\sqrt{n}} \right| \left| \mathbb{E}_{\mathcal{D}_A} h(\mathbf{x}) \sigma'(\mathbf{w}^\top \mathbf{x} + b) - \varphi(\mathbf{w}, b) \mathbb{E}_{\mathcal{D}_A} h(\mathbf{x}) \right|$$

$$\le \frac{C_1 \sqrt{n}}{\sqrt{|J|}} \le \frac{\sqrt{3} C_1 \sqrt{n}}{\sqrt{(n-k)}} \le \sqrt{6} C_1$$

Now, let $\alpha_i = \varphi(\mathbf{w}, b)$ and recall that $\varphi(\mathbf{w}, b) = \mathbb{P}\left[ \frac{k}{\sqrt{n}} < \sum_{j \in J} w_j x_j + b < 6 - \frac{k}{\sqrt{n}} \right]$, and since $\frac{k}{\sqrt{n}} \le \frac{1}{8k}$ and $b = \frac{1}{8k}$ we have:

$$\alpha_i \ge \mathbb{P}\left[ 0 \le \sum_{j \in J} w_j x_j < 5 \right] = \frac{1}{2} - \mathbb{P}\left[ \sum_{j \in J} w_j x_j > 5 \right]$$

From Markov's inequality we have: $\mathbb{P}\left[ \left| \sum_{j \in J} w_j x_j \right| > 5 \right] \le \frac{1}{5^2}$. And from symmetry we get that $\mathbb{P}\left[ \sum_{j \in J} w_j x_j > 5 \right] \le \frac{1}{2 \cdot 5^2} \le \frac{1}{4}$, and so $\alpha_i \ge \frac{1}{4}$. Finally, for every $j \notin A$, using Lemma 7 we get:

$$\left| w_{i,j}^{(1)} \right| = \left| u_i^{(0)} \right| \left| \frac{1}{2} \mathbb{E}_{\mathcal{D}_A^{(1)}} x_j f_A(\mathbf{x}) \sigma'(\mathbf{w}^\top \mathbf{x} + b) + \frac{1}{2} \mathbb{E}_{\mathcal{D}_A^{(2)}} x_j f_A(\mathbf{x}) \sigma'(\mathbf{w}^\top \mathbf{x} + b) \right|$$

$$= \frac{n}{2k} \left| \mathbb{E}_{\mathcal{D}_A^{(1)}} x_j f_A(\mathbf{x}) \sigma'(\mathbf{w}^\top \mathbf{x}) \right| \le 7 \frac{n(n-1)}{\sqrt{k}} \sqrt{\frac{1}{\binom{n-1}{k}}}$$

$$\le \frac{7}{n-1}(n-1)^3 \sqrt{\frac{1}{\binom{n-1}{6}}} \le \frac{7}{n-1}(n-1)^3 \sqrt{\frac{6^6}{(n-1)^6}} \le 7 \cdot \frac{6^3}{n-1}$$

$\square$

*Proof.* of Lemma 10. Denote $u^* = -\frac{bn}{\alpha r}$, and let $\epsilon' = \frac{bn}{\alpha |r| k} \epsilon$. Notice that $|u^*| \le \frac{n}{2k}$ so $[u^* - \epsilon', u^* + \epsilon'] \subset [-\frac{n}{k}, \frac{n}{k}]$. Therefore, we get that $\mathbb{P}[|u - u^*| \le \epsilon'] = \frac{\epsilon' k}{n} = \frac{b \epsilon}{\alpha |r|} \ge \frac{1}{8k^2} \epsilon$. Notice that:

$$\phi_r(z) = \frac{|r|}{b} \sigma\left( \frac{\alpha}{n} u^* z + b \right)$$

And therefore:

$$\left|\frac{|r|}{b}\sigma(\frac{\alpha}{n}uz+b)-\phi_r(z)\right|=\frac{|r|}{b}\left|\sigma(\frac{\alpha}{n}uz+b)-\sigma(\frac{\alpha}{n}u^*z+b)\right|\le\frac{|rz|\alpha}{bn}|u-u^*|\le\frac{r\alpha k}{bn}\epsilon'=\epsilon$$

$\square$

*Proof.* of Lemma 11. From Lemma 9, with probability at least $\frac{1}{14\sqrt{k}}$ over the choice of $\mathbf{w}_i^{(0)}$, we have that: $\max_{j\in A}\left|w_{i,j}^{(0)}-\frac{\alpha_i}{\sqrt{n}}u_i^{(0)}\right|\le C_1$, $\max_{j\notin A}\left|w_{i,j}^{(0)}-\frac{\alpha_i}{\sqrt{n}}u_i^{(0)}\right|\le\frac{C_2}{n-1}$ and $\left|b^{(1)}-b^{(0)}\right|\le\frac{C_3}{\sqrt{n}}$ for some universal constants $C_1,C_2,C_3$, and some $\alpha_i\in\left[\frac{1}{4},1\right]$ depending only on $\mathbf{w}_i^{(0)}$. From Lemma 10, with probability at least $\frac{\epsilon}{8k^2}$ over the choice of $u_i^{(0)}$ (and independently of the choice of $\mathbf{w}_i^{(0)}$), we have $\left|\frac{|r|}{b_i^{(0)}}\sigma(\frac{\alpha}{n}u_i^{(0)}z+b_i^{(0)})-\phi_r(z)\right|\le\epsilon$ for every $z\in[-k,k]$.

Assume the results of both lemmas hold, which happens with probability at least $\frac{\epsilon}{112k^{2.5}}$. Now, fix some $\mathbf{x}\in\mathcal{X}$ and let $z=\sqrt{n}\sum_{j\in A}x_j\in[-k,k]$. Then we have:

$$\begin{aligned}
\left|\widehat{\psi}_i(\mathbf{x})-\psi_r(\mathbf{x})\right|&=\left|\frac{|r|}{b_i^{(0)}}\sigma\left(\left\langle\mathbf{w}_i^{(1)},\mathbf{x}\right\rangle+b_i^{(1)}\right)-\sigma\left(-\operatorname{sign}(r)z+|r|\right)\right|\\
&=\frac{|r|}{b_i^{(0)}}\left|\sigma\left(\left\langle\mathbf{w}_i^{(1)},\mathbf{x}\right\rangle+b_i^{(1)}\right)-\sigma\left(\frac{\alpha_i}{n}u_i^{(0)}z+b_i^{(0)}\right)\right|\\
&\quad+\left|\frac{|r|}{b_i^{(0)}}\sigma\left(\frac{\alpha_i}{n}u_i^{(0)}z+b_i^{(0)}\right)-\sigma\left(-\operatorname{sign}(r)z+|r|\right)\right|
\end{aligned}$$

From the result of Lemma 9:

$$\begin{aligned}
&\left|\sigma\left(\left\langle\mathbf{w}_i^{(1)},\mathbf{x}\right\rangle+b_i^{(1)}\right)-\sigma\left(\frac{\alpha_i}{\sqrt{n}}u_i^{(0)}z+b_i^{(0)}\right)\right|\\
&\le\left|\left\langle\mathbf{w}_i^{(1)},\mathbf{x}\right\rangle+b_i^{(1)}-\frac{\alpha_i}{n}u_i^{(0)}z+b_i^{(0)}\right|\\
&\le\left|\left\langle\mathbf{w}_i^{(1)},\mathbf{x}\right\rangle-\frac{\alpha_i}{\sqrt{n}}u_i^{(0)}\sum_{j\in A}x_j\right|+\left|b_i^{(1)}-b_i^{(0)}\right|\\
&\le\sum_{j\in A}\left|w_{i,j}^{(1)}x_j-\frac{\alpha_i}{\sqrt{n}}u_i^{(0)}x_j\right|+\sum_{j\notin A}\left|w_{i,j}^{(1)}x_j\right|+\left|b_i^{(1)}-b_i^{(0)}\right|\\
&\le\frac{kC_1}{\sqrt{n}}+\frac{C_2}{\sqrt{n}}+\frac{C_3}{\sqrt{n}}
\end{aligned}$$

Using the result of Lemma 10 we get that:

$$\left|\widehat{\psi}_i(\mathbf{x})-\psi_r(\mathbf{x})\right|\le\frac{|r|}{b_i^{(0)}}\left(\frac{C_1k+C_2+C_3}{\sqrt{n}}\right)+\epsilon\le\frac{C_4k^4}{\sqrt{n}}+\epsilon$$

For some universal constant $C_4$. Using the assumption on $k$ concludes the proof. $\square$

*Proof.* of Lemma 12. Fix some $r\in\{-k,-k+2,\ldots,k-2,k\}$. Let $\epsilon=\frac{1}{10k}$, and from Lemma 11, with probability at least $\frac{1}{1120k^{3.5}}$ over the choice of $\mathbf{w}_i^{(0)},u_i^{(0)}$ we have:

$$\left|\widehat{\psi}_i(\mathbf{x})-\psi_r(\mathbf{x})\right|\le\frac{1}{10k}$$

Assume $q\ge2\cdot1120^2k^7\log(\frac{k+1}{\delta})$. Denote $I_r=\{i\in[q]:\left|\widehat{\psi}_i(\mathbf{x})-\psi_r(\mathbf{x})\right|\le\frac{1}{10k}\}$. Denote $p:=\frac{1}{1120k^{3.5}}$, and using Hoeffding's inequality, with probability at least $1-\exp\{-\frac{p^2}{2}q\}\ge1-\frac{\delta}{k+1}$

we have $|I_r| \geq \frac{p}{2}q$. Therefore, using the union bound we get that with probability at least $1 - \delta$, for every $r \in \{-k, -k+2, \ldots, k-2, k\}$ we have $|I_r| \geq \frac{p}{2}q$. Let $J_r \subset I_r$ be some subset of size $|J_r| = \frac{p}{2}q$. Define:

$$v_r = \begin{cases} 1 & |r| = k \\ 2.5 & |r| = 1 \\ 2 & 1 < |r| < k \end{cases}$$

Observe that $\sum_r (-1)^{\frac{k-r}{2}} v_r \psi_r(\mathbf{x}) = f_A(\mathbf{x})$. Therefore, we have that:

$$\left| \frac{2}{pq} \sum_r \sum_{i \in J_r} (-1)^{\frac{k-r}{2}} v_r \widehat{\psi}_i(\mathbf{x}) - f_A(\mathbf{x}) \right| = \left| \frac{2}{pq} \sum_r \sum_{i \in J_r} (-1)^{\frac{k-r}{2}} v_r \widehat{\psi}_i(\mathbf{x}) - \sum_r (-1)^{\frac{k-r}{2}} v_r \psi_r(\mathbf{x}) \right|$$

$$\leq \frac{2}{pq} \sum_r \sum_{i \in J_r} |v_r| \left| \widehat{\psi}_i(\mathbf{x}) - \psi_r(\mathbf{x}) \right|$$

$$\leq 2.5(k+1)\frac{1}{10k} \leq \frac{1}{2}$$

Define:

$$u_i^* = \begin{cases} (-1)^{\frac{k-r}{2}} \frac{2v_r|r|}{pqb_i^{(0)}} & \exists r \ s.t \ i \in J_r \\ 0 & o.w \end{cases}$$

Now, we have $|u_i| \leq \frac{2}{pq}10(k+1)k \leq \frac{Bk^{5.5}}{q}$ where $B$ is a universal constant. Therefore, we get that $\|\mathbf{u}^*\| \leq \sqrt{\frac{q(k+1)}{2240k^2}} \cdot \frac{Bk^{5.5}}{q} = B' \frac{k^5}{\sqrt{q}}$. From what we showed, such $\mathbf{u}^*$ achieves the required. $\square$

*Proof.* of Theorem 13. We follow an analysis similar to [23]. Let $R_t(\theta) = \sum_{i=1}^t \langle \theta, \nabla f_i \rangle + \frac{1}{2\eta} \|\theta\|^2$, and notice that $\arg\min_\theta R_t = -\eta \sum_{i=1}^t \nabla f_i = \theta_{t+1} - \theta_1$. We show by induction that for every $\theta^*$ we have:

$$\sum_{t=1}^T \langle \theta_{t+1} - \theta_1, \nabla f_t(\theta_t) \rangle \leq \sum_{t=1}^T \langle \theta^*, \nabla f_t(\theta_t) \rangle + \frac{1}{2\eta} \|\theta^*\|^2 = R_T(\theta^*) \tag{5}$$

First, we have:

$$\langle \theta_2 - \theta_1, \nabla f_t(\theta_t) \rangle \leq R_1(\theta_2 - \theta_1) \leq R_1(\theta^*)$$

since $\theta_2 - \theta_1$ minimizes $R_1$. Now, assume the above is true for $T - 1$, then we have:

$$\sum_{t=1}^{T-1} \langle \theta_{t+1} - \theta_1, \nabla f_t(\theta_t) \rangle \leq \sum_{t=1}^{T-1} \langle \theta_{T+1} - \theta_1, \nabla f_t(\theta_t) \rangle$$

And by adding $\langle \theta_{T+1} - \theta_1, \nabla f_T(\theta_T) \rangle$ to both sides we get:

$$\sum_{t=1}^T \langle \theta_{t+1} - \theta_1, \nabla f_t(\theta_t) \rangle \leq \sum_{t=1}^T \langle \theta_{T+1} - \theta_1, \nabla f_t(\theta_t) \rangle \leq R_T(\theta_{T+1} - \theta_1) \leq R_T(\theta^*)$$

Now, from (5) we get that:

$$\sum_{t=1}^T \langle \theta_t - \theta_1, \nabla f_t(\theta_t) \rangle - R_T(\theta^*) \leq \sum_{t=1}^T \langle \theta_t - \theta_1, \nabla f_t(\theta_t) \rangle - \sum_{t=1}^T \langle \theta_{t+1} - \theta_1, \nabla f_t(\theta_t) \rangle$$

$$= \sum_{t=1}^T \langle \theta_t - \theta_{t+1}, \nabla f_t(\theta_t) \rangle = \eta \sum_{t=1}^T \|\nabla f_t(\theta_t)\|^2$$

Using Cauchy-Schwartz inequality and rearranging the above yields:

$$\sum_{t=1}^T \langle \theta_t - \theta^*, \nabla f_t(\theta_t) \rangle \leq \frac{1}{2\eta} \|\theta^*\|^2 + \|\theta_1\| \sum_{t=1}^T \|\nabla f_t(\theta_t)\| + \eta \sum_{t=1}^T \|\nabla f_t(\theta_t)\|^2$$

Finally, from convexity of $f_t$ we get:

$$\sum_{t=1}^{T}(f_t(\theta_t) - f_t(\theta^*)) \le \sum_{t=1}^{T}\langle \theta_t - \theta^*, \nabla f_t\rangle \le \frac{1}{2\eta}\|\theta^*\|^2 + \|\theta_1\|\sum_{t=1}^{T}\|\nabla f_t(\theta_t)\| + \eta\sum_{t=1}^{T}\|\nabla f_t(\theta_t)\|^2$$

$\square$

*Proof.* of Lemma 14. W.l.o.g., assume $A = [k]$. Denote $I_{even} := \{\mathbf{z} \in \{\pm\frac{1}{\sqrt{n}}\}^k \ : \ \prod_i z_i > 0\}$ and $I_{odd} := \{\mathbf{z} \in \{\pm\frac{1}{\sqrt{n}}\}^k \ : \ \prod_i z_i < 0\}$. Notice that since $k$ is odd, we have $I_{odd} = -I_{even}$. From the symmetric initialization we have $g^{(0)} \equiv 0$. By definition of the gradient-updates, we have:

$$
\begin{aligned}
u_i^{(1)} &= u_i^{(0)} - \eta_1\left(\frac{\partial}{\partial u_i}L_{\mathcal{D}}(g^{(0)}) + \lambda_1\frac{\partial}{\partial u_i^{(0)}}R(g^{(0)})\right) \\
&= u_i^{(0)} - \mathbb{E}\left[\ell'(f_A(\mathbf{x}), g^{(0)}(\mathbf{x}))\frac{\partial}{\partial u_i}g^{(0)}(\mathbf{x})\right] - \frac{1}{2}\frac{\partial}{\partial u_i^{(0)}}R(g^{(0)}) \\
&= -\mathbb{E}\left[f_A(\mathbf{x})\sigma(\langle \mathbf{w}_i^{(0)}, \mathbf{x}\rangle + b)\right] \\
&= -\sum_{\mathbf{z}\in I_{even}}\mathbb{E}\left[\sigma(\langle \mathbf{w}_i^{(0)}, \mathbf{x}\rangle + b)|\mathbf{x}_{1\ldots k} = \mathbf{z}\right]\mathbb{P}\left[\mathbf{x}_{1\ldots k} = \mathbf{z}\right] \\
&\quad + \sum_{\mathbf{z}\in I_{even}}\mathbb{E}\left[\sigma(\langle \mathbf{w}_i^{(0)}, \mathbf{x}\rangle + b)|\mathbf{x}_{1\ldots k} - \mathbf{z}\right]\mathbb{P}\left[\mathbf{x}_{1\ldots k} = -\mathbf{z}\right]
\end{aligned}
$$

Since by definition of the distribution $\mathcal{D}_A$ we have $\mathbb{P}\left[\mathbf{x}_{1\ldots k} = \mathbf{z}\right] = \mathbb{P}\left[\mathbf{x}_{1\ldots k} = -\mathbf{z}\right]$, we get that:

$$
\begin{aligned}
u_i^{(1)} &= \sum_{\mathbf{z}\in I_{even}}\mathbb{P}\left[\mathbf{x}_{1\ldots k} = \mathbf{z}\right]\mathbb{E}\left[\sigma(\sum_{j=1}^{k}w_{i,j}^{(0)}z_j + \sum_{j=k+1}^{n}w_{i,j}^{(0)}x_j + b)\right] \\
&\quad - \sum_{\mathbf{z}\in I_{even}}\mathbb{P}\left[\mathbf{x}_{1\ldots k} = \mathbf{z}\right]\mathbb{E}\left[\sigma(-\sum_{j=1}^{k}w_{i,j}^{(0)}z_j + \sum_{j=k+1}^{n}w_{i,j}^{(0)}x_j + b)\right]
\end{aligned}
$$

And since $\sigma$ is 1-Lipschitz we get:

$$\left|u_i^{(1)}\right| \le \sum_{\mathbf{z}\in I_{even}}\mathbb{P}\left[\mathbf{x}_{1\ldots k} = \mathbf{z}\right]2\left|\sum_{j=1}^{k}w_{i,j}^{(0)}z_j\right| \le \frac{k}{\sqrt{n}}2\sum_{\mathbf{z}\in I_{even}}\mathbb{P}\left[\mathbf{x}_{1\ldots k} = \mathbf{z}\right] = \frac{k}{\sqrt{n}}$$

Where we use the fact that $\sigma$ is 1-Lipschitz. $\square$

*Proof.* of Lemma 15. From Lemma 14 we have that $\left|u_i^{(1)}\right| \le \frac{k}{\sqrt{n}}$. For every $t > 1$:

$$
\begin{aligned}
\left|u_i^{(t)}\right| &= \left|u_i^{(t-1)} - \eta\frac{\partial}{\partial u_i}L_{\mathcal{D}}(g^{(t-1)}) - \eta\lambda\frac{\partial}{\partial u_i}R(g^{(t-1)})\right| \\
&= \left|u_i^{(t-1)} - \eta\mathbb{E}\left[\ell'(f_A(\mathbf{x}), g^{(t-1)}(\mathbf{x}))f_A(\mathbf{x})\sigma(\langle \mathbf{w}_i^{(t-1)}, \mathbf{x}\rangle + b_i^{(t-1)})\right] - 2\eta\lambda u_i^{(t-1)}\right| \\
&\le \left|(1 - 2\eta\lambda)u_i^{(t-1)} - 6\eta\right| \le \left|u_i^{(t-1)}\right| + 6\eta \le \cdots \le \left|u_i^{(1)}\right| + 6\eta(t-1) \le 6\eta t + \frac{k}{\sqrt{n}}
\end{aligned}
$$

Now, using the above we get that:

$$
\begin{aligned}
\left\| \mathbf{w}_i^{(t)} - \mathbf{w}_i^{(1)} \right\| &= \left\| \mathbf{w}_i^{(t)} - \eta \frac{\partial}{\partial w_i} L_{\mathcal{D}}(g^{(t-1)}) - \eta \lambda \frac{\partial}{\partial w_i} R(g^{(t-1)}) \right\| \\
&= \left\| \mathbf{w}_i^{(t-1)} - \mathbf{w}_i^{(1)} - \eta \mathbb{E}\left[ \ell'(f_A(\mathbf{x}), g^{(t-1)}(\mathbf{x})) u_i^{(t-1)} \sigma'(\mathbf{w}^\top \mathbf{x} + b)\mathbf{x} \right] - 2\eta\lambda \mathbf{w}_i^{(t-1)} \right\| \\
&\leq \left\| \mathbf{w}_i^{(t-1)} - \mathbf{w}_i^{(1)} - 2\eta\lambda \mathbf{w}_i^{(t-1)} \right\| + \eta \left| u_i^{(t-1)} \right| \\
&\leq (1 - 2\eta\lambda) \left\| \mathbf{w}_i^{(t-1)} - \mathbf{w}_i^{(1)} \right\| + 2\eta\lambda \left\| \mathbf{w}_i^{(1)} \right\| + 6\eta^2 t + \eta \frac{k}{\sqrt{n}} \\
&\leq \left\| \mathbf{w}_i^{(t-1)} - \mathbf{w}_i^{(1)} \right\| + 2\eta\lambda \frac{n}{k} + 6\eta^2 t + \eta \frac{k}{\sqrt{n}} \leq \cdots \leq 2\eta t \lambda \frac{n}{k} + 6\eta^2 t^2 + \eta t \frac{k}{\sqrt{n}}
\end{aligned}
$$

Where we use the fact that:

$$
\left\| \mathbf{w}_i^{(1)} \right\| = \left\| \mathbb{E}\left[ \ell'(f_A(\mathbf{x}), g^{(0)}(\mathbf{x})) u_i^{(0)} \sigma'(\mathbf{w}^\top \mathbf{x} + b)\mathbf{x} \right] \right\| \leq \left| u_i^{(0)} \right| \leq \frac{n}{k}
$$

Finally, for the bias we get:

$$
\begin{aligned}
\left| b_i^{(t)} - b_i^{(1)} \right| &= \left| b_i^{(t)} - \eta \frac{\partial}{\partial b_i} L_{\mathcal{D}}(g^{(t-1)}) \right| \\
&= \left| b_i^{(t-1)} - b_i^{(1)} - \eta \mathbb{E}\left[ \ell'(f_A(\mathbf{x}), g^{(t-1)}(\mathbf{x})) u_i^{(t-1)} \sigma'(\mathbf{w}^\top \mathbf{x} + b) \right] \right| \\
&\leq \left| b_i^{(t-1)} - b_i^{(1)} \right| + \eta \left| u_i^{(t-1)} \right| \\
&\leq \left| b_i^{(t-1)} - b_i^{(1)} \right| + 6\eta^2 t + \eta \frac{k}{\sqrt{n}} \leq \cdots \leq 6\eta^2 t^2 + \eta t \frac{k}{\sqrt{n}}
\end{aligned}
$$

$\square$

*Proof.* of Lemma 16. Denote the support of $u^*$ by $I := \{i \in [2q] \ : \ u_i^* \neq 0\}$. Then we have:

$$
\begin{aligned}
\left| \ell(g_{\mathbf{u}^*}^{(t)}(\mathbf{x}), y) - \ell(g_{\mathbf{u}^*}^{(1)}(\mathbf{x}), y) \right| &\leq \left| g_{\mathbf{u}^*}^{(t)}(\mathbf{x}) - g_{\mathbf{u}^*}^{(1)}(\mathbf{x}) \right| \\
&= \left| \sum_{i \in I} u_i^* \left( \sigma\left( \left\langle \mathbf{w}_i^{(t)}, \mathbf{x} \right\rangle + b_i^{(t)} \right) - \sigma\left( \left\langle \mathbf{w}_i^{(1)}, \mathbf{x} \right\rangle + b_i^{(1)} \right) \right) \right| \\
&\leq \| \mathbf{u}^* \|_2 \sqrt{|I|} \left| \sigma\left( \left\langle \mathbf{w}_i^{(t)}, \mathbf{x} \right\rangle + b_i^{(t)} \right) - \sigma\left( \left\langle \mathbf{w}_i^{(1)}, \mathbf{x} \right\rangle + b_i^{(1)} \right) \right| \\
&\leq \| \mathbf{u}^* \|_2 \sqrt{|I|} \left( \left| \left\langle \mathbf{w}_i^{(t)}, \mathbf{x} \right\rangle - \left\langle \mathbf{w}_i^{(1)}, \mathbf{x} \right\rangle \right| + \left| b_i^{(t)} - b_i^{(1)} \right| \right) \\
&\leq \| \mathbf{u}^* \|_2 \sqrt{|I|} \left( \left\| \mathbf{w}_i^{(t)} - \mathbf{w}_i^{(1)} \right\| + \left| b_i^{(t)} - b_i^{(1)} \right| \right)
\end{aligned}
$$

Using Lemma 15 we get:

$$
\left| \ell(g_{\mathbf{u}^*}^{(t)}(\mathbf{x}), y) - \ell(g_{\mathbf{u}^*}^{(1)}(\mathbf{x}), y) \right| \leq \| \mathbf{u}^* \|_2 \sqrt{|I|} \left( 12\eta^2 t^2 + 2\eta t \frac{k}{\sqrt{n}} + 2\eta t \lambda \frac{n}{k} \right)
$$

$\square$