[Reviews · NeurIPS 2020]

Review 1

Summary and Contributions: This is a theoretical work that gives an example of a class of (distribution,hypothesis)-pairs which: (i) can be learnt efficiently using a 1-layer neural net with random initialization and gradient descent; yet (ii) under any embedding, any linear function obtaining vanishing loss requires exponentially large weights. The construction goes roughly as follows: the hypotheses (parity on a subset $A$ of the coordinates) is quite hard given uniform samples. The distribution has 50% weight on uniform samples, and another 50% on samples that are particularly easy for the given hypothesis (all coordinates in $A$ have identical signs). Since the embedding cannot depend on the distribution, it has no chance of being useful on the hard part for a typical hypothesis. For the learning algorithm, the algorithm first uses the easy half to learn to assign high, positive weights to coordinates in $A$. Then the algorithm needs to learn the parity on A, which by inclusion-exclusion can be expressed efficiently as a weighted sum of the nodes with high weight on A. On the other hand, any fixed embedding cannot assign sufficient weight to the interaction between coordinates in A since there are n-choose-k candidate subsets.

Strengths: The submission makes progress on a very central question: how come neural networks perform better than classical models in machine learning. Even though the example is obviously contrived, maybe we can extract some useful insights: the neural network is winning because it first learns the useful features (using the first half of the distribution), and then learns from them (using the second half of the distribution).

Weaknesses: Maybe it would have been more exciting to separate deep from shallow. (But I agree with authors that this is a good problem for future research.)

Correctness: I believe so

Clarity: For the most part yes. The verbal explanations between technical lemma statements are very helpful.

Relation to Prior Work: Yes

Reproducibility: Yes

Additional Feedback: Line 138: be [an] odd numbers Line 144: w_i should be w_j Line 153: relevant coordinate*s* Line 196: pf -> of Lemma 7: Shouldn’t it be about w^(1)-u^(0) rather than w^(0)-u^(0)? Also, for j \notin A, don’t you want to prove that w^(1)_j is close to 0 rather than to u^(0)? I can’t follow the derivation between lines 373 and 374, and also the one after line 375.


Review 2

Summary and Contributions: This paper studies the benefit of learning pairities using one-hidden-layer neural networks under favorable distributional assumptions. It shows two main results: 1. Any linear classifier with over a fixed feature representation will have a high (\Omega(1)) approximation error on the parity learning problem under the uniform distribution over instance domain {-1/sqrt{n}, 1/\sqrt{n}}^n (D_A^1 in the paper), unless the feature representation is exponentially-many dimensional. 2. Under favorable distributional assumptions (the underlying distribution is a mixture of uniform distribution D_A^1 and a distribution that reveals information about the target parity function D_A^2), training a wide one-hidden-layer neural network by (population-level) gradient descent with certain step size schedule can find a network that has a low error rate.

Strengths: - Motivates the necessity of neural work training by formally showing limitations of using a fixed representation - Have a detailed convergence analysis of stochastic gradient descent procedure. At a high level the plan is clear: (1) they show that there are many neurons initialized such that its weight w satisfies \sum_{i \in A} w_i = 0 (A is the support of the parity); (2) applying a step of gradient descent, these neurons will have coefficients that induces majority-like structure over A; (3) these neuron have good diversity in its offset values, so there is some way of linearly combining them that yields a good parity function (4) online gradient descent can help approximate the target function. - The MNIST experiment is surprisingly good to me, showing the power of neural network in learning parities. On the other hand, I don't know if the data distributions in the concatenated MNIST data align with with ones assumed in this paper. Can the authors clarify?

Weaknesses: - The biggest weakness I can see is the strong distributional assumption employed in this paper. Specifically, the unlabeled distribution D_A^2 depends specifically on the support of parity A. I cannot think of real-world examples where such an assumption holds. If the authors can elaborate on how the techniques in this paper can be used to analyze neural net learning parities under other distributional assumptions that would be great.

Correctness: Most of the claims look correct to me - I have a few minor technical questions: - Is it easy to extend the paper's reasoning to learning with parities with over an even number of coordinates? - It looks like the current argument relies on the regularizer's gradient cancels with the initialized vector. How sensitive is the analysis to the first step size and the regularizer strength? - In the proof of Lemma 6, why can we ignore \sum_{j \in A} w_j x_j in w^T x? Also, why do we need to consider the segment [-k/sqrt{n}, 0]? - In page 372, did you use Holder's inequality and the fact that \ell' is bounded by 1?

Clarity: Yes, except some clarity issues in the proof. See above.

Relation to Prior Work: Yes.

Reproducibility: No

Additional Feedback: - The final error bound of SGD given by Theorem 2 has a term qk/sqrt(n) that does not go to zero as the number of iterations T and the number of neurons q increases. Can the authors clarify if this is fundamental? - I am confused why in Theorem 11, the third term in the right hand side (\| \theta_1 \| 1/T \sum_{t=1}^T \| \nabla f_t(\theta_t) \|) is present? I think standard results such as [22] you cite does not have this term at all. -- After rebuttal -- With the help of the meta-reviewer, my technical concerns are resolved. In addition, given that learning parity is hard under the uniform distribution using statistical queries, I now appreciate why the paper imposes a strong assumption on the unlabeled distribution. Nevertheless, the gap between the theory and the experiment in this paper is still a bit large in this respect...


Review 3

Summary and Contributions: This paper studies the problem of learning k-parities using depth 2 neural networks, under a specific simple distribution. Theorem 1 shows that k-parity cannot be approximated by linear models and Theorem 2 shows that GD learns such models.

Strengths: This is the first result to my knowledge showing a separation between linear / kernel methods and NN. The proof of the learning result is nontrivial and entails a delicate analysis of the gradient of the loss, which must separately address coordinates on versus off the support of the parity. The lower bound for linear methods is fairly standard and unsurprising The theoretical results are supported empirically by experiments on MNIST data. The result is highly relevant to the NeurIPS community and is a nice step forward.

Weaknesses: It would be useful to clearly explain the role of having a mixture distribution over purely uniform and all + or all - within set A. This eventually becomes evident but I think it would add clarity to say a word about this earlier. Also it would be nice to remark to what extent you expect a similar statement to hold for general distributions. It's clear that this distribution is fairly special in creating nonzero gradient in parity coordinates. Another question, why is it important to initialize w in {-1,0,1}^n rather than {-1,1}^n?

Correctness: As far as I can tell, the mathematical arguments are correct.

Clarity: The paper is quite well-written. There are fairly frequent small grammatical errors, but they do not detract significantly from the presentation. Lemma 4 statement can be clarified ("for all j...") What's going on with Lemma 7? Shouldn't some of these entries have a 1 in the superscript? Also you wanted for j\notin A that w_{ij} is small, not close to u_i, so line 163 seems to need some correcting.

Relation to Prior Work: It would be useful to contrast carefully with the results of https://arxiv.org/abs/1812.06369. Clearly the distribution plays an important role!

Reproducibility: Yes

Additional Feedback:


Review 4

Summary and Contributions: The authors give the proof that a linear model with fixed features cannot learn the parity problem (xor problem or if the sum of digits is even) while a 2-layer neural network can. The separation is of exponential nature. ---- read the reviews and the rebuttal and want to keep my score ----

Strengths: * The authors extend the theory by showing that a broader function family (namely learning parity) distinguishes random features+linear model from end-to-end trained 2-layer or deeper neural networks by an exponential learning factor. * Neat experiment to show it in practice.

Weaknesses: [25] shows a similar result, admitting with a smaller function family.

Correctness: I did not read the proofs. The experimental section is sound.

Clarity: yes.

Relation to Prior Work: Minor comment: "An empirical study on the properties of random bases for kernel methods" treats the topic from a practical standpoint of view.

Reproducibility: Yes

Additional Feedback: I'm afraid, the paper escaped my initial view; I don't have enough mathematical/statistical learning theory background to assess the paper well. From an educated view after reading it twice it seems written well and coherent. I did not go through the proofs in detail.

[Author Response · NeurIPS 2020]

We thank the reviewers for their very positive and encouraging feedback.

Regarding the main comments raised by the reviewers:

- The strong distributional assumptions: we believe similar techniques can be used for showing similar results under more general families of distributions, a problem that we leave for future work. Specifically, the key property that we used for showing our positive result is the fact that the gradient-descent draws the weights of the irrelevant bits to zero, a behavior that can be observed in other distributions as well. We focused on this choice of distribution to make the analysis simpler, and since this setting suffices to demonstrate the separation between neural networks and linear classes, which is the main goal of the paper.

- The choice of a parities over odd number of bits is important since we assume that, in half of the distribution, the parity bits are either all 1 or all -1. We believe this assumption can be removed when considering different distributional assumptions.

- We introduced the regularization term to simplify the theoretical analysis, but we believe similar results can be shown in different settings as well.

We will fix and address the additional comments and suggestions raised by the reviewers in the final version of the paper.

[Meta-Review · NeurIPS 2020]

This is a very nice paper showing that gradient descent on a carefully regularized hinge loss, two layer network can learn certain sparse parity problems with sample complexity in a certain sense exponentially better than linear methods (including RKHS and NTK after converting to finite width). The reviews and discussions were all eventually in favor, and I personally really enjoyed this paper and also its proof, which I studied in some detail, and plan to investigate even more. --- Minor comments. (a) I think it would be useful to include a corollary with an explicit choice or range of q, which combines both the upper and lower bound into a single clear compelling separation. Personally, to understand things, I chose q= k^18 and n>= k^40, which I think is kindof implied as natural by the upper bound (it makes the bound like 1/k). While these exponents are large, I think it would help the story for many readers. You can even write "poly"... (b) I think you can spend some time/space explaining the distributional assumption, and what breaks down if you remove it. (c) There was a reviewer question about a proof detail, which you ignored in your (short) rebuttal. Ultimately, it was me who checked the proof and explained the step to the reviewer, and not another reviewer. Please in the future clarify such issues, it is IMO one of the good things about these rebuttals, dealing with actual technical questions.